# Textile Triboelectric Nanogenerators with Diverse 3D-Spacer Fabrics for Improved Output Voltage

**Dae-Hyeon Kwon [1], Jin-Hyuk Kwon [1], Jaebum Jeong [2], Youngju Lee [3,4], Swarup Biswas [4], Dong-Wook Lee [3], Sohee Lee [5,*], Jin-Hyuk Bae [1,6,*] and Hyeok Kim [4,*]**

[1]  School of Electronic and Electrical Engineering, Kyungpook National University, 80 Daehakro, Bukgu, Daegu 41566, Korea; kdh0663@naver.com (D.-H.K.); rnjs3055@naver.com (J.-H.K.)

[2]  Nano Materials & Nano Technology Center, Korea Institute of Ceramic Engineering and Technology (KICET), Jinju 52851, Korea; iriver0208@naver.com

[3]  Applied Robot R&D Department, Korea Institute of Industrial Technology (KITECH), Ansan 15588, Korea; dydwn0603@naver.com (Y.L.); dwlee@kitech.re.kr (D.-W.L.)

[4]  School of Electrical and Computer Engineering, Institute of Information Technology, University of Seoul, 163 Seoulsiripdaero, Dongdaemun-gu, Seoul 02504, Korea; Biswas.swarup1988@gmail.com

[5]  Department of Clothing and Textiles, Research Institute of Natural Science, Gyeongsang National University, 501 Jinjudaero, Jinju, Gyeongsangnamdo 52828, Korea

[6]  School of Electronic Engineering, Kyungpook National University, Daegu 41566, Korea

*  Correspondence: sohee.lee@gnu.ac.kr (S.L.); jhbae@ee.knu.ac.kr (J.-H.B.); hyeok.kim@uos.ac.kr (H.K.); Tel.: +82-55-772-1458 (S.L.); +82-53-950-7222 (J.-H.B.); +82-2-6490-2354 (H.K.)

**Abstract:** Electrically superior triboelectric nanogenerators (TENG) using 3D fabric and PDMS show great application potential for biokinetic energy harvesting and multifunctional self-power devices. In this study, TENG with fabric-PDMS-fabric structure was produced using various 3D fabrics and PDMS. The peak-to-peak output voltage of various 3D fabrics was compared. The output voltage changes due to structure and vertical fibers. Also, the coefficient of surface friction between the PDMS and the fabric improves the output voltage. TENG using different 3D-spacer polymeric fabrics showed different maximum peak-to-peak output voltage performance. It is attributed to the stiffness, lateral elasticity and 3D morphology of the fabrics. It is considered that those factors including stiffness, lateral elasticity and 3D morphology influence the densities in vertical and lateral fiber to fiber interaction.

**Keywords:** 3D fabric; triboelectric nanogenerator; knitting structure; output voltage.

## 1. Introduction

As securing sustainable and eco-friendly energy becomes an issue around the world, energy harvesting technology that harvests electric energy from energy sources that are wasted or consumed in everyday life such as light, heat, and vibration is in the spotlight [1–4]. In addition, fiber has been used by mankind for a long time and is essential for everyday life. Bendable, portable, and foldable, you can adapt it to your daily activities [5,6]. Versatility and wearability, portable electronics are evolving rapidly and are a great advantage for communications, personal health care and environmental monitoring. Therefore, among many energy harvesting devices, the flexible triboelectric nanogenerator (TENG) using fibers received great attention because it has a wearable energy harvesting function that acquires and stores energy generated from the interaction between the environment around the human body and human activities [7–9]. These can be easily used energy sources for wearable electronics, portable electronics.

However, many studies require a complex and expensive process for high-performance TENG using fibers [10–12]. In addition, TENG, which can be worn on the human body, must be comfortable and harmless to the wearer, and must conveniently harvest energy from body movements. In order to overcome these problems, research on

various fibers is also necessary, and TENG performance improvement and device structure studies using fibers are also essential [13–15]. TENG, which uses a typical 2D fabric structure, has many problems. This is because the TENG's performance is lowered due to the limited contact area and surface triboelectricity.

For this reason, in order to introduce various fibers that provide comfortable elasticity to the human body and to improve the output voltage of TENG using fibers, this study proposes 3D fabrics that combine single jersey, honeycomb, and tricot structures. Type 1 fabric consists of a single jersey with foil finishing-honeycomb and 100% polyester (PE). Type 2 fabric consists of single jersey-single jersey and 92% PE and 8% spandex (SP). Type 3 fabric is composed of honeycomb-tricot structure and 100% PE. Also, the output voltage and current of each 3D fabric-TENG is compared. In addition, in this study, we used a Fabric-PDMS-fabric (FPF) structure that shows excellent output voltage by a high coefficient of surface friction between the PDMS and the fabric. This research is expected to contribute to the commercialization of next-generation wearable electronic devices and sleep monitoring sensors.

## 2. Materials and Methods

We fabricated TENG from three 3D fabrics and polydimethylsiloxane (PDMS). Tribo-negative and tribo-positive surfaces maximize the improvement of output voltage of TENG due to high potential different in TENG. Therefore, in this work, we introduced PDMS (tribo-negative) into fabrics (tribo-positive). Figure 1 shows photographs of the front and back sides of the 3D fabrics used in this study, respectively. In Figure 1a, Type 1 fabric is made of 100% PE, with a single jersey with foil finish on the front and honeycomb on the back. In Figure 1b, Type 2 fabric consists of 92% PE and 8% SP, and has a single jersey structure on both the front and back sides. In Figure 1c, Type 3 fabric is made of 100% PE, and the front is a honeycomb structure and the back is a tricot structure. Figure 2 shows the schematic diagram and stacking method of TENG of FPF structure. In order to manufacture the TENG of the FPF structure, PDMS belonging to the middle layer was produced. PDMS was prepared using Sylgard-184 from an elastomeric PDMS kit by Dow-Corning. A 10:1 by weight base/curing agent mixture was stirred more than 10 min. After mixing, the mixture was placed in vacuum desiccators to remove trapped air bubbles. The mixture was then poured on a silicon wafer and cured at 100 C for 1 h. Each layer between PDMS and fabric was stacked with copper tape electrodes. All TENGs are made 8 cm × 8 cm. During the testing of TENGs we fixed the applied force = value at 10 N and measurement frequency at 8 Hz.

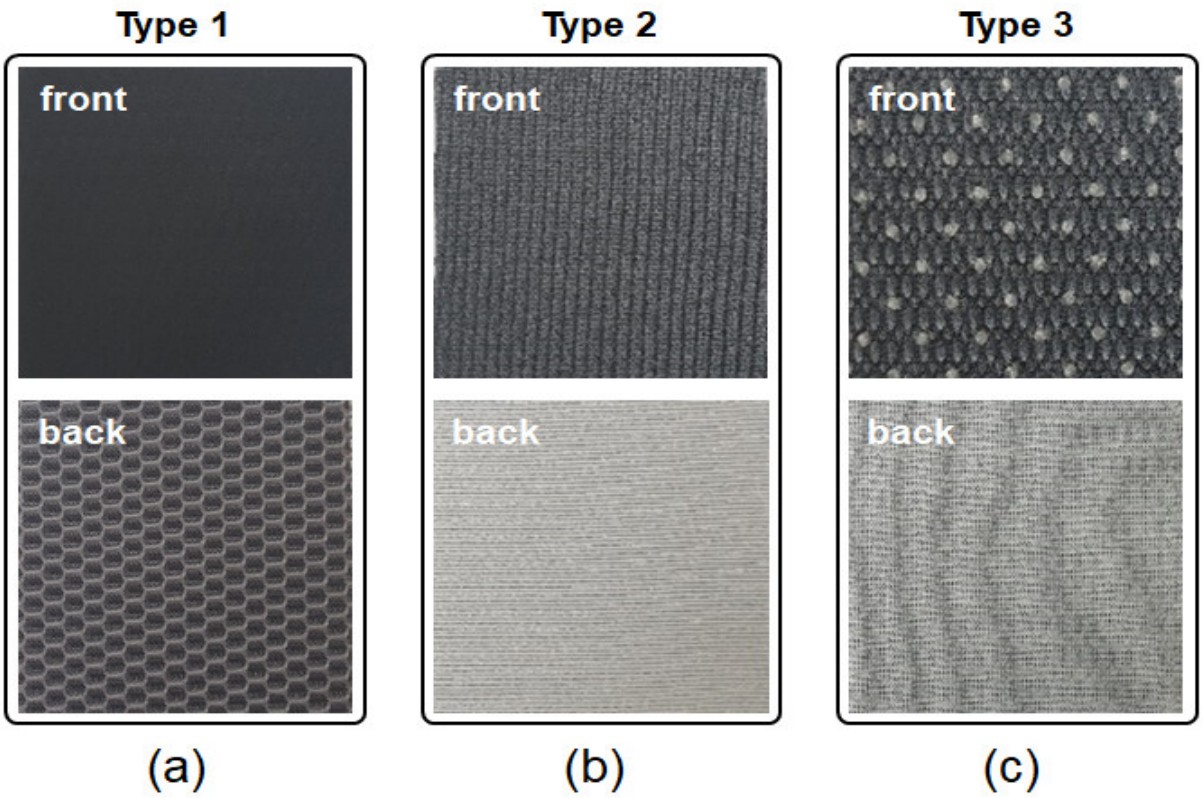

**Figure 1.** (**a**) The pictures of the front and back of Type 1 fabric; (**b**) The pictures of the front and back of Type 2 fabric; (**c**) The pictures of the front and back of Type 3 fabric.

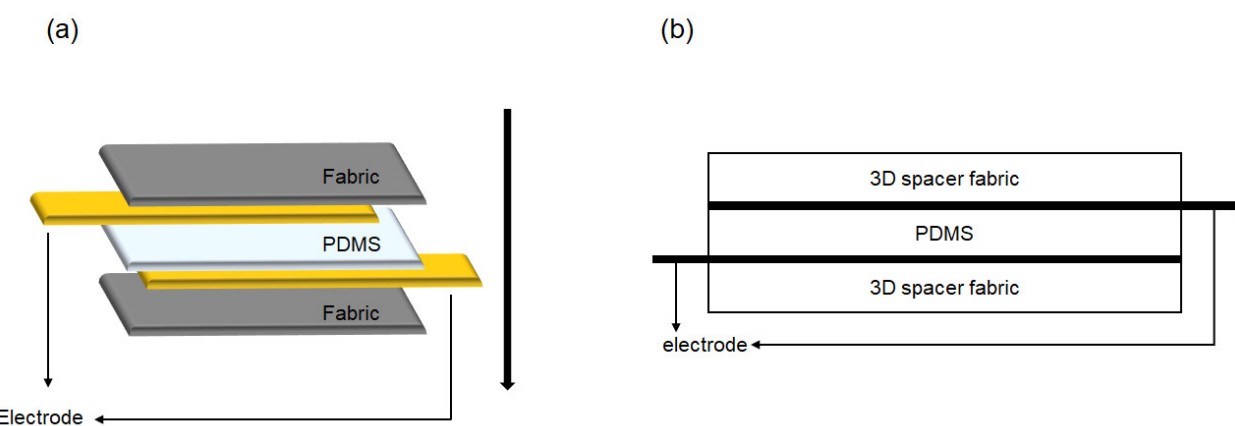

**Figure 2.** (**a**) The stacking order of the constituent fabric, PDMS, and electrode layers; (**b**) the TENG cross section.

### 3. Results and Discussion

Figure 2 shows the schematic diagram of TENG with fabric-PDMS-fabric (FPF) structure. In Figure 2a, you can see that the fabric-PDMS-fabric is stacked in the order. Also, there is an electrode between the fabric and PDMS. The reason for using PDMS is that the high coefficient of surface friction between PDMS and fabric shows good output voltage. Figure 2b shows a cross section of the finished TENG.

Figure 3 shows the scanning electron microscopy (SEM) images of each 3D-spacer fabric. Figure 3a–d shows the SEM images of Type 1 fabric. Figure 3a shows the front surface of Type 1 fabric. Figure 3a shows a single jersey structure with a foil finish [16]. Foil finishing makes less stretchable compared to single jersey construction without foil

finish. Figure 3b shows the back surface of Type 1 fabric. Figure 3b clearly shows the surface of the hexagonal honeycomb structure [17]. Figure 3c exhibits the (100) cross section of Type 1 fabric. Figure 3d shows the (010) cross section of Type 1 fabric. Figure 3c shows the vertically placed fibers between the top and bottom of the fabric. Figure 3d shows the vertically placed fibers between the top and bottom of the fabric. It is twisted on both sides, unlike Figure 3c. Vertical fibers maximize the triboelectric effect even with small deformations. Figure 3e–h shows the SEM images of Type 2 fabric. Figure 3e shows the front surface of Type 2 fabric. Figure 3f shows the back surface of Type 2 fabric. Figure 3e,f show the same structure as a single jersey structure. Since the single jersey structure has low elasticity in the width direction, the elasticity in the width direction was supplemented by mixing spandex in Type 2 fabric [18]. Figure 3g shows the (100) cross section of Type 2 fabric. Figure 3h shows the (010) cross section of Type 2 fabric. Since this is not a honeycomb structure, the number of vertical fibers is less than that of Type 1 fabric and Type 3 fabric. Also, unlike the honeycomb structure, when viewed from (100) and (010), the vertical fibers lie in only one direction. Figure 3i–l shows the SEM images of Type 3 fabric. Figure 3i shows the front surface of Type 3 fabric. Figure 3g clearly shows the surface of the hexagonal honeycomb structure, as in Figure 3b. Figure 3j shows the back surface of Type 3 fabric. Figure 3j shows the tricot structure. The tricot structure has excellent elasticity [19]. Figure 3k shows the (100) cross section of Type 3 fabric. Figure 3l shows the (010) cross section of Type 3 fabric. As shown in Figure 3c,d of Type 1 fabric, Figure 3k,l show vertically placed fibers of Type 3 fabric. Like Type 1, it is twisted on both sides.

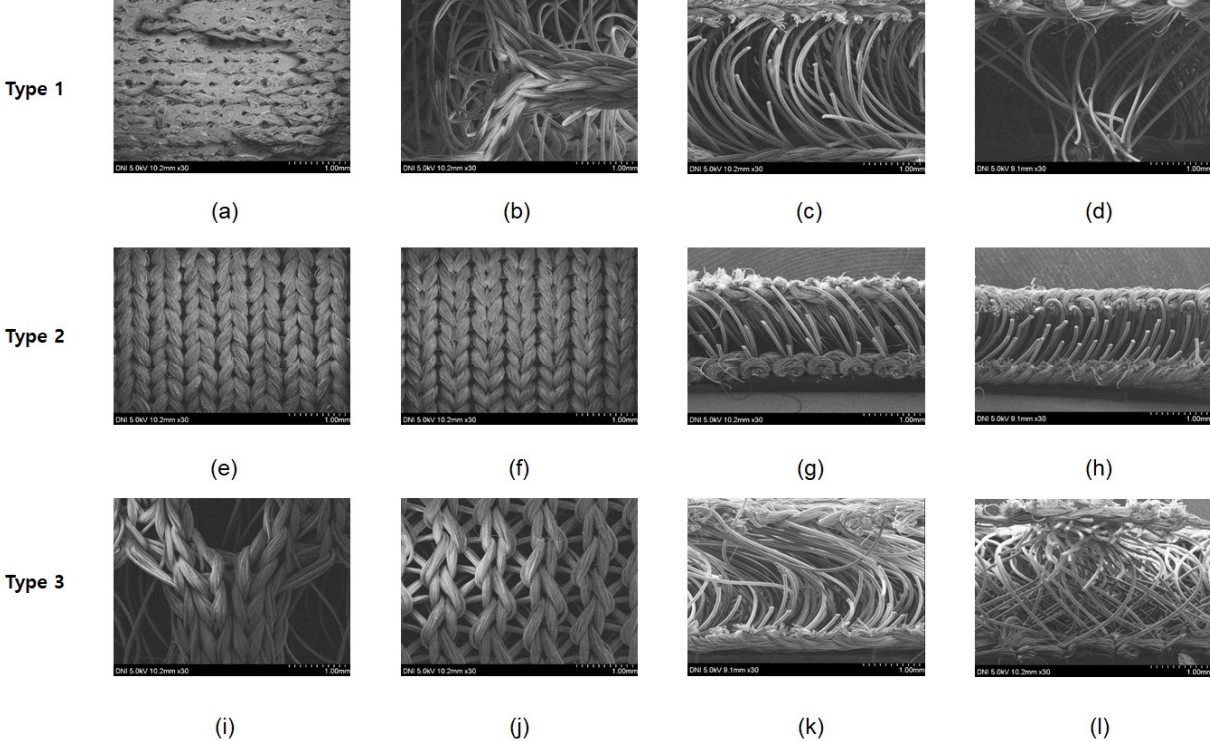

**Figure 3.** (**a**) The front surface of Type 1 fabric. (**b**) The back surface of Type 1 fabric; (**c**) The (100) cross section of Type 1 fabric; (**d**) The (010) cross section of Type 1 fabric; (**e**) The front surface of Type 2 fabric; (**f**) The back surface of Type 2 fabric; (**g**) The (100) cross section of Type 2 fabric; (**h**) The (010) cross section of Type 2 fabric; (**i**) The front surface of Type 3 fabric; (**j**) The back surface of Type 3 fabric; (**k**) The (100) cross section of Type 3 fabric; (**l**) The (010) cross section of Type 3 fabric.

Figure 4 shows the output voltage and current of three 3D-spacer fabric TENGs using the FPF structure over time. In Figure 4a, the peak-to-peak output voltage of Type 1 fabric TENG is 30 V, showing the lowest output voltage characteristics. The back side of Type 1

is a honeycomb structure with a hexagonal honeycomb structure. The honeycomb structure promotes deformation and increases the contact area and friction when the two layers experience contact and separation [17]. However, the Type 1 fabric with a single jersey structure with foil finish has a stiff surface, which has low elasticity which results low output voltage. In Figure 4b, the peak-to-peak output voltage of Type 2 fabric TENG is 44.7 V. Both the front and the back have a single jersey structure, so it has lower elasticity compared to the honeycomb structure, but by mixing 8% SP, the elasticity in the width direction is increased to show a compliant output voltage. The SP is mainly composed of polyurethane, which can increase the original length and maintains its original elasticity [18]. In Figure 4c, the peak-to-peak output voltage of Type 3 fabric TENG is shows the highest output voltage characteristic at 50.3V. As with Type 1 fabric, since the back side has a honeycomb structure, the contact area is large and the friction coefficient is high, so it shows excellent output voltage characteristics. In addition, the tricot structure on the front side has good lateral elasticity, which induces high frictional charge generation, showing the highest peak-to-peak output voltage.

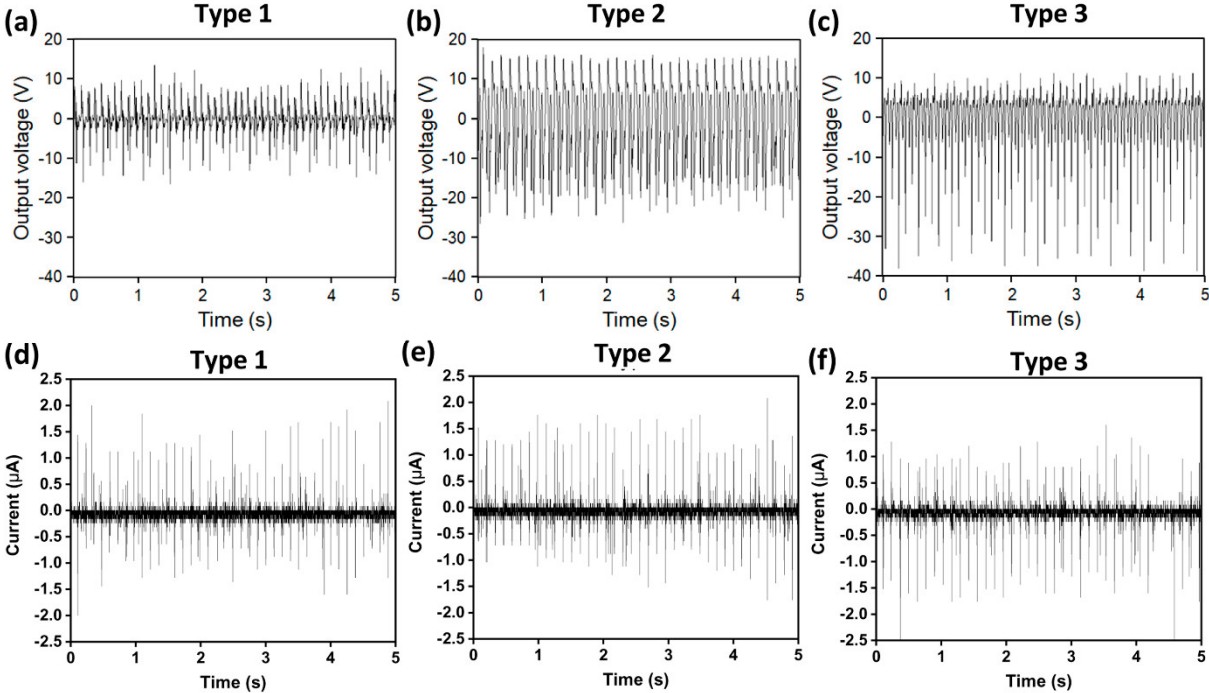

**Figure 4.** The output voltages (**a**–**c**) and currents (**d**–**f**) of Type 1–3 TENGs.

The Figure 4d–f are representing the output currents of Type 1, Type 2, and Type 3 TENG respectively. Here it is very interesting to observe that, there is a very little variation between the output currents characteristics curves of different TENGs and more important variation comes from the output voltage of each device. Since the differentiation of potential is proportional to the surface charge density. Therefore, a massive variation is found in output voltages not in currents. Figure 5 is a graph comparing the peak-to-peak output voltages of three types of TENG produced for this study. The output voltage of Type 3 TENG with honeycomb (front), tricot (back) structure improves the triboelectric effect according to the structure and shows higher peak-to-peak output voltage than Type 2 TENG and Type 1 TENG This shows that even in TENG with the same structure, the output voltage can be improved depending on what kind of fabric is used.

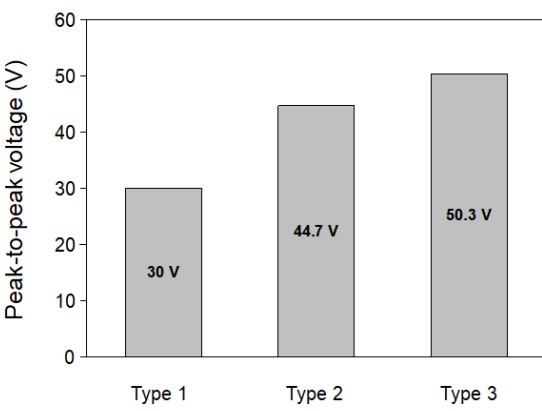

**Figure 5.** The maximum peak-to-peak output voltage of Type 1–3 TENGs.

Besides the other reasons, there might be some effect of the variation of surface roughness of the fabrics onto the variations in output voltage values of different TENGs. To check this, we further characterized the surface roughness of different fabrics. As the fabric is made from thin threads, it is almost impossible to estimate surface roughness of the fabric by analyzing AFM image. The side view SEM images (Figure 3) of the samples suggest that the surface roughness of the fabrics are in mm scale. As it is impossible to estimate the surface roughness values of different fabrics from AFM measurement, here we have tried to utilize contact angle measurement for estimating the surface roughness factor (Table 1) of different fabrics indirectly [20]. For estimation of surface roughness factor, we have employed Cassie–Baxter equation Equation (1), which is most appropriate for these types of systems [20].

$$Cos\,\theta_A = r_f\left(Cos\,\theta_Y + 1\right) - 1 \tag{1}$$

where $\theta_A$ is apparent contact angle, $\theta_Y$ is the contact angle value estimated by Young's relation, and $r_f$ is the surface roughness factor (ratio of true area of the solid surface to the apparent area. i.e., $r_f$ =1 stands for smooth surface).

From the Table 1 it can be observed that the back surface of Type 3 fabric has most rough surface (having minimum surface roughness factor value). Along with the other factors such as good lateral elasticity, higher surface roughness may also be another reason (a little influence [18]) behind the achievement of highest output voltage from the Type 3 fabric based TENG.

**Table 1.** Surface roughness factor of top and bottom surface of Type 1, Type 2, and Type 3 fabrics estimated from contact angle (with water) values.

| Fabric | Surface | Apparent contac angle (°) | Young's contact angle (°) | Roughness factor ($r_f$) |
|---|---|---|---|---|
| Type 1 | Front | 84.37 | 82.4 | 0.97 |
| | Back | - | - | - |
| Type 2 | Front | 80.21 | 77.25 | 0.96 |
| | Back | 79.63 | 75.24 | 0.93 |
| Type 3 | Front | 76.99 | 70.58 | 0.93 |
| | Back | 72.72 | 64.62 | 0.90 |



## 4. Conclusions

We proposed three 3D-spacer fabrics to improve the output voltage of TENG and compared the output voltage characteristics. The three 3D-spacer fabrics exhibited changes in output voltage due to surface friction, structure, and vertical fibers. Type 1 fabric TENG, which has a honeycomb structure but uses a single jersey structure with a foil finish, has low elasticity, resulting in the lowest output voltage of 30 V. By adding SP to the Type 2 fabric with only a single jersey structure, the elasticity in the width direction was increased, and the Type 2 fabric TENG showed an output voltage of 44.7 V. Type 3 fabric TENG using the tricot structure and honeycomb structure further increased the transverse stretch, resulting in the highest output voltage of 50.3 V We expect this study to produce high-efficiency TENG in an inexpensive and simple process. In addition, it is expected to contribute to the commercialization of next-generation wearable electronic devices and sleep monitoring sensors.

**Author Contributions:** Conceptualization, D.-H.K., S.L., J.-H.B. and H.K.; data curation, J.J. and Y.L.; writing—original draft preparation, D.-H.K. and J.-H.K.; writing—review and editing, J.-H.K., J.-H.B. and H.K.; project administration, D.-H.K., J-H.K., J.J., D.-W.L., Y.L., S.B., S.L., J.-H.B. and H.K. All authors have read and agreed to the published version of the manuscript.

**Funding:** This work was supported by the 2021 Research Fund of the University of Seoul and by Basic Science Research Program through the National Research Foundation of Korea (NRF) funded by the Ministry of Science and ICT (2021R1A2C1011429) and by Basic Science Research Program through the National Research Foundation of Korea (NRF) funded by Ministry of Education, Science and Technology (NRF-2018R1D1A3B07049551).

**Conflicts of Interest:** The authors declare no conflict of interest.

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
