# Peer review of "Textile Triboelectric Nanogenerators with Diverse 3D-Spacer Fabrics for Improved Output Voltage"

_electronics, doi:10.3390/electronics10080937_

Round 1

Reviewer 1 Report

This paper reports the triboelectric nanogenerators featuring various 3D fabrics as a contact layer. Testing triboelectrification of 3D fabrics is worth trying but the experiment should be designed systematically. Therefore, I do not recommend this paper for further processing, and below are some comments that can be improved in the paper.

1) Triboelectricity is mainly dependent on both surface roughness and material’s intrinsic properties. Therefore, if the authors would like to look at the effect of surface roughness, they should use the same materials.

2) Triboelectricity is also varied by the applied force so that the authors should mention it in the papers.

3) The surface roughness should be quantitatively characterized.

4) The charge generation is a quite important parameter to evaluate the performance of TENGs so that the authors should measure the short-circuit current of TENGs.

5) I don’t quite get the reason why the authors used the PDMS layer in the middle.

Reviewer 2 Report

The results obtained must be interpreted at least minimally relevant.

For example the data represented in figure 4 can be analyzed spectrally (FFT for example).

The results presented in figure 5 must be commented and correlated etc.

At least indicative values of electric current (or electric charges) is need to be specified, not just the electric potential etc.

The important elements must be highlighted and compared more relevant with some from the literature.

Reviewer 3 Report

This manuscript reports minor delta improvement over previously published paper. This is a classic case of "Salami publishing". Even SEM images seem to have been reused.

Kim, D.K.; Jeong, J.B.; Lim, K.; Ko, J.; Lang, P.; Choi, M.; Lee, S.; Bae, J.H.; Kim, H. Improved output voltage of a nonagenera-tor with 3D fabric. Journal of nanoscience and nanotechnology. 2020, 20, 4666-4670. 

Author Response

SEM images were obtained for this study and have not been reused for other papers or research data.
This study is a follow-up to the reference that you commented. However, the reference is a study showing the difference between 2D and 3D fabric in TENG, and this study is a study investigating the electrical characteristics of TENG according to the difference in 3D fabric structure. The two studies are correlated, but there are clear differences.

Round 2

Reviewer 1 Report

I pointed out the experiment should be designed systematically in the previous review. Simple revision of several sentences and addition of a few data can not be considered as a major change in the experimental design. Therefore, I do not recommend this paper for further processing. 

Author Response

Thank you so much for helping with the review.

concerning surface roughness, we've done our best.

This research itself, investigation on surface roughness for 3D fabric, is a new topic of research.

Therefore, the authors would like to continue to study this work and

have a chance to submit a new article to Electronics as the consecutive work of this work.

Reviewer 2 Report

..

Author Response

Thank you so much for helping with the review.